# Imaging low-mass planets within the habitable zone of α Centauri

K. Wagner [1,2✉], A. Boehle [3], P. Pathak [4], M. Kasper[4], R. Arsenault[4], G. Jakob[4], U. Käufl [4], S. Leveratto[4], A.-L. Maire[5], E. Pantin[6], R. Siebenmorgen [4], G. Zins[4], O. Absil [5], N. Ageorges[7], D. Apai[1,2,8], A. Carlotti[9], É. Choquet [10], C. Delacroix [5], K. Dohlen [10], P. Duhoux[4], P. Forsberg[11], E. Fuenteseca[4], S. Gutruf[7], O. Guyon[1,12,13,14], E. Huby[15], D. Kampf[7], M. Karlsson [11], P. Kervella [15], J.-P. Kirchbauer[4], P. Klupar[13], J. Kolb[4], D. Mawet[16], M. N'Diaye [17], G. Orban de Xivry [5], S. P. Quanz [3], A. Reutlinger[7], G. Ruane[16,18], M. Riquelme[4], C. Soenke[4], M. Sterzik [4], A. Vigan [10] & T. de Zeeuw[4,19,20]

Giant exoplanets on wide orbits have been directly imaged around young stars. If the thermal background in the mid-infrared can be mitigated, then exoplanets with lower masses can also be imaged. Here we present a ground-based mid-infrared observing approach that enables imaging low-mass temperate exoplanets around nearby stars, and in particular within the closest stellar system, α Centauri. Based on 75–80% of the best quality images from 100 h of cumulative observations, we demonstrate sensitivity to warm sub-Neptune-sized planets throughout much of the habitable zone of α Centauri A. This is an order of magnitude more sensitive than state-of-the-art exoplanet imaging mass detection limits. We also discuss a possible exoplanet or exozodiacal disk detection around α Centauri A. However, an instrumental artifact of unknown origin cannot be ruled out. These results demonstrate the feasibility of imaging rocky habitable-zone exoplanets with current and upcoming telescopes.

[1] Dept. of Astronomy and Steward Observatory, University of Arizona, Tucson, AZ, USA. [2] NASA Nexus for Exoplanet System Science, Earths in Other Solar Systems Team, Tucson, AZ, USA. [3] Institute for Particle Physics and Astrophysics, ETH Zurich, Zürich, Switzerland. [4] European Southern Observatory, Garching bei München, Germany. [5] STAR Institute, Université de Liège, Liège, Belgium. [6] AIM, CEA, CNRS, Université Paris-Saclay, Université Paris Diderot, Sorbonne Paris Cité, Gif-sur-Yvette, France. [7] Kampf Telescope Optics, München, Germany. [8] Lunar and Planetary Laboratory, University of Arizona, Tucson, AZ, USA. [9] Univ. Grenoble Alpes, CNRS, IPAG, Grenoble, France. [10] Aix Marseille Univ, CNRS, CNES, LAM, Marseille, France. [11] Department of Materials Science and Engineering, Ångström Laboratory, Uppsala University, Uppsala, Sweden. [12] Subaru Telescope, National Astronomical Observatory of Japan, National Institutes of Natural Sciences (NINS), Hilo, HI, USA. [13] The Breakthrough Initiatives, NASA Research Park, Moffett Field, CA, USA. [14] James C. Wyant College of Optical Sciences, University of Arizona, Tucson, AZ, USA. [15] LESIA, Observatoire de Paris, Meudon, France. [16] California Institute of Technology, Pasadena, CA, USA. [17] Université Côte d'Azur, Observatoire de la Côte d'Azur, CNRS, Laboratoire Lagrange, Nice, France. [18] Jet Propulsion Laboratory, California Institute of Technology, Pasadena, CA, USA. [19] Sterrewacht Leiden, Leiden University, Leiden, The Netherlands. [20] Max Planck Institute for Extraterrestrial Physics, Garching, Germany. ✉email: kevinwagner@email.arizona.edu

A primary pursuit of modern astronomy is the search for worlds that are potentially similar to Earth. Such worlds would help us to understand the context of our own planet and would themselves become targets of searches for life beyond the solar system (e.g., refs. [1–3]). Meanwhile, giant exoplanets have been imaged on wide orbits—enabling direct studies of their orbits and atmospheres (e.g., refs. [4–6]). To enable finding and exploring potentially Earth-like planets, exoplanet imaging capabilities are progressing towards lower-mass planets in the habitable zones of nearby stars (e.g., refs. [7–9]). In this context, habitable refers to the possibility of a planet with a broadly Earth-like atmosphere to host liquid water on its surface.

The nearest stellar system, α Centauri, is among the best-suited for imaging habitable-zone exoplanets (e.g., refs. [10–12]). The primary components α Centauri A and B are similar in mass and temperature to the Sun, and their habitable zones are at separations of about one au (see ref. [13] and Fig. 1). At the system's distance of 1.3 pc, these physical separations correspond to angular separations of about one arcsecond, which can be resolved with existing 8-m-class telescopes. However, no planets are currently known to orbit either star. Measurements of the stars' radial velocity (RV) trends[14] exclude planets more massive than $M\sin i \geq 53$ Earth-masses ($M_\oplus$) in the habitable zone of α Centauri A, and $\geq 8.4\,M_\oplus$ for α Centauri B. Lower-mass planets could still be present and dynamically stable (e.g., ref. [15]). The tertiary M-dwarf component of the system, Proxima Centauri, also hosts at least two planets more massive than Earth[16,17] that were discovered through the star's RV variations.

Conventional exoplanet imaging studies (e.g., refs. [18–20]) have operated at wavelengths of $\lambda \leq 5\,\mu m$, in which the background noise is relatively low (i.e., the sensitivity is dominated by residual starlight), but in which temperate planets are faint compared to their peak emission in the mid-infrared ($\lambda \sim 10$–$20\,\mu m$). The exoplanets that have been imaged are young super-Jovian planets on wide orbits ($a > 10$ au) with temperatures of $\sim 10^3$ K (e.g., refs. [18–21]). Their high temperatures are a remnant of formation and reflect their youth ($\sim 1$–100 Myr, compared to the Gyr ages of typical stars). Imaging potentially habitable planets will require imaging colder exoplanets on shorter orbits around mature stars. This leads to an opportunity in the mid-infrared ($\sim 10\,\mu m$), in which temperate planets are brightest. However, mid-infrared imaging introduces significant challenges. These are primarily related to the much higher thermal background—that saturates even sub-second exposures—and also the $\sim 2$–$5\times$ coarser spatial resolution due to the diffraction limit scaling with wavelength. With current state-of-the-art telescopes, mid-infrared imaging can resolve the habitable zones

of roughly a dozen nearby stars, but it remains to be shown whether sensitivity to detect low-mass planets can be achieved.

In this work, we present the results of the New Earths in the α Centauri Region (NEAR:[22,23]) experiment. As part of Breakthrough Watch[24], NEAR aims to demonstrate experimental technologies and techniques to facilitate directly imaging low-mass habitable-zone exoplanets. Specifically, NEAR aims to demonstrate that low-mass exoplanets can be imaged in a practical, but unprecedented amount of observing time (~100 h) by conducting a direct imaging search for habitable-zone exoplanets within the nearest stellar system, α Centauri. We describe the NEAR campaign, an analysis of the sensitivity to habitable-zone exoplanets, and an assessment of a candidate detection. Finally, we discuss possibilities for imaging rocky habitable-zone exoplanets around α Centauri and other nearby stars with these techniques.

## Results

NEAR explored an instrumental setup and observing strategy designed to push the capabilities of ground-based exoplanet imaging toward mid-infrared wavelengths of ~10 μm. The mid-infrared camera (VISIR:[25]) on the Very Large Telescope (VLT) was upgraded for NEAR to implement several new technologies, such as a mid-infrared optimized annular groove phase mask (AGPM:[26]) coronagraph[27,28] and shaped-pupil mask[29,30] to suppress the starlight. VISIR was moved to unit telescope 4 (UT4/Yepun) of the VLT, which is equipped with a deformable secondary mirror (DSM,[31]). The DSM is a crucial element of NEAR's strategy, as it enabled performing adaptive optics (AO) without additional non-cryogenic corrective optics whose thermal emission would contribute to the total background. The DSM was also utilized to alternate the position of the binary behind the coronagraph (chopping) with a frequency of ~10 Hz, while pausing the AO state during the transition between stars. Subtracting the images taken during alternate positions removed much of the background–partly including the contribution from the annular groove phase mask[32]. With this strategy, planets around either star would appear centered on the coronagraph, with planets orbiting α Centauri A appearing as positive point sources and planets orbiting α Centauri B appearing as negative sources. Additional details about the experiment design can be found in refs. [22,23], and methods, instrumental setup and observing Strategy.

α Centauri was observed from 2019 May 23 to 2019 June 11. The nightly conditions can be found in Supplementary Table 1. An additional night of data was taken on 2019 June 27. Enough

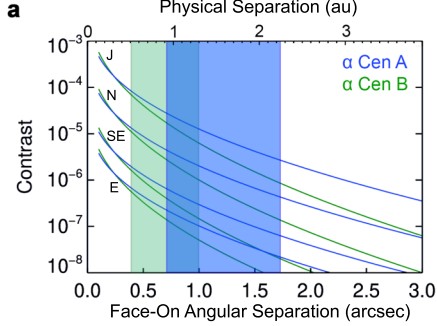
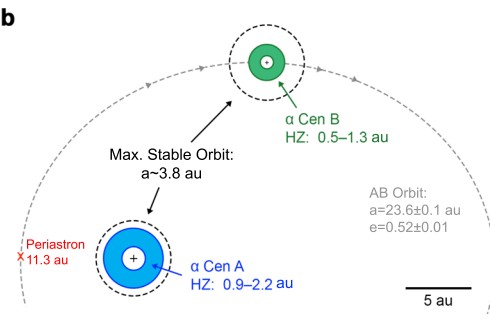

**Fig. 1 Simulated planet brightness and basic properties of the α Centauri system. a** *N*-band (10–12.5 μm) contrast vs. angular separation of planets around α Centauri A (blue) and B (green), assuming face-on circular orbits, a Bond albedo of 0.3 and internal heating that provides an additional 10% of the planets' equilibrium temperatures. The curves correspond from bottom to top to planetary radii equivalent to that of Earth, a Super-Earth (1.7 × Earth's radius, $R_\oplus$), Neptune, and Jupiter. The blue and green shaded regions show the location of the classical habitable zones around α Centauri A and B, respectively[13]. **b** Diagram of the orbital properties and approximate habitable zones of the α Centauri AB system. Note that this diagram does not show the 79° inclination of the orbit as seen from Earth, or the tertiary dwarf star, Proxima Centauri, at ~$10^4$ au.

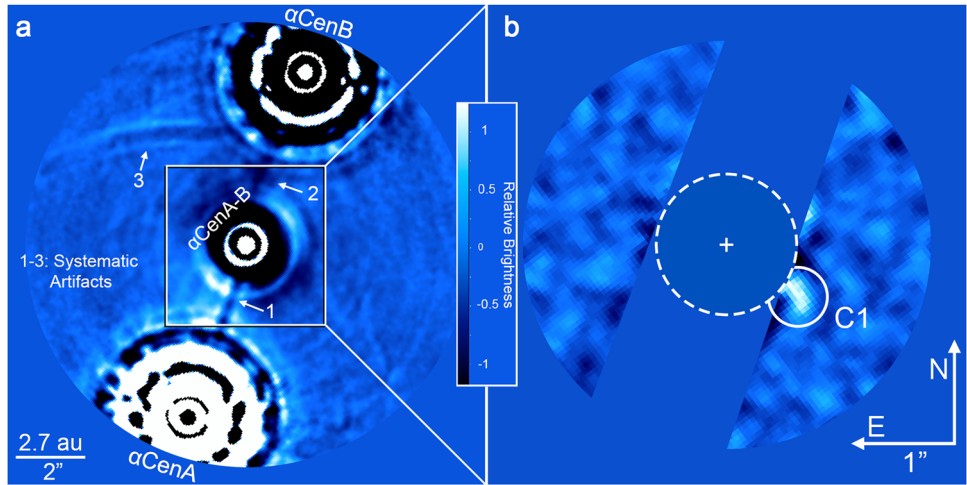

**Fig. 2 Mid-infrared images of α Centauri. a** high-pass filtered image without PSF subtraction or artifact removal. The α Centauri B on-coronagraph images have been subtracted from the α Centauri A on-coronagraph images, resulting in a central residual and two off-axis PSFs to the SE and NW of α Centauri A and B, respectively. Systematic artifacts labeled 1–3 correspond to detector persistence from α Centauri A, α Centauri B, and an optical ghost of α Centauri A. **b** Zoom-in on the inner regions following artifact removal and PSF subtraction. Regions impacted by detector persistence are masked for clarity. The approximate inner edge of the habitable zone of α Centauri A[13] is indicated by the dashed circle. A candidate detection is labeled as 'C1'.

time separates the collection of these data from the initial observations such that orbital motion complicates combining these with the rest of the data[33,34]. However, this extra night provides a useful astrometric check for brighter planet candidates. Including these data, we collected a cumulative exposure time of approximately 100 h, out of which 23 h were not used because of mediocre data quality as a result of high sky background, coronagraph misalignment, or AO problems. The remaining 76.9 h of good quality data were used for the subsequent analysis. Further details can be found in methods, data reduction and processing.

Following stellar point spread function (PSF) subtraction, the brightest features in the images are systematic artifacts (see Fig. 2 and Supplementary Fig. 1). The most significant artifact is detector persistence accumulated during the chopping sequence. The second most significant artifacts that appear in the final images are negative arcs due to optical ghosts (reflections) of the off-axis PSF of α Centauri A that are introduced by the dichroic beam-splitter and spectral filter. These artifacts limit the overall image sensitivity and increase the false positive probability within specific regions. To improve the overall image sensitivity, we modeled and subtracted each of the known artifacts (see Supplementary Fig. 1). The full-frame image prior to artifact subtraction and a zoom-in on the habitable zone following artifact subtraction are shown in Fig. 2.

An Earth-sized planet at a separation of $\rho \sim 1.2''$ ($a \sim 1.5$ au) around α Centauri A would appear with $\sim 10^{-7}$ contrast at $\lambda \sim 10$ μm (see Fig. 1, and also ref. [22]). A super-Earth ($R \sim 1.7\,R_\oplus$) at the inner edge of the habitable zone ($\rho \sim 0.8''$) would appear with a contrast of $\sim 10^{-6}$. As an assessment of the detector's fundamental sensitivity limit (in the absence of residual stellar flux and spatially correlated noise), we examined the standard deviation of pixel intensities within 1.2 $\lambda/D$ ($\sim 0.35''$, or $\sim 8$ pixels) in diameter in a region of the detector far from α Centauri A and B (see supplementary methods, background-limited sensitivity). We found this value (multiplied by the square root of the number of pixels contained within the aperture) to be $\sim 1.67 \times 10^{-7}$ contrast with respect to α Centauri A, or about $\sim 22$ μJy. The pixel-to-pixel noise increases toward the glow of the AGPM[32]. At $1''$ separation, the standard deviation of pixel intensities is roughly doubled by the glow.

For an empirical assessment of the detection sensitivity to planets (i.e., point sources), we performed numerous simulated point-source injection and retrieval tests via forwarding modeling injected signals of planetary-brightness throughout the data processing (see supplementary methods, simulated planet injection and retrieval tests). Injected sources are identified in the images at about an order of magnitude above the (1-σ) background noise, or $\sim 2$–$3 \times 10^{-6}$ contrast to α Centauri A (see Figs. 3, 4). Over much of the habitable zone of α Centauri A, the image sensitivity is sufficient to detect $R \geq 9\,R_\oplus$ planets in radiative thermal equilibrium (i.e., Saturn-sized, with $T \sim 300$ K), and smaller planets with additional heat or low Bond albedos. Within α Centauri B's habitable zone, the images reach sensitivities to detect Jupiter-sized planets ($R \sim 11\,R_\oplus$) with a small amount of additional heat ($T = 1.1 \times T_{eq}$) or a low Bond albedo.

We converted the sensitivity analysis into a completion estimate using a Monte Carlo simulation to draw randomly sampled orbital parameters (see Fig. 5 and supplementary methods, completeness analysis for details). With no prior orbital constraints, the NEAR data reach a maximum completeness of $\sim 80\%$ for Jovian-sized planets, or $\sim 85\%$ for slightly larger (i.e., inflated) planets. The maximum completeness is less than unity since the projected separation can be smaller than a given semi-major axis, or behind the persistence stripes. At the extreme end of the detection limits, there is a $\sim 1$–$10\%$ chance of detecting a warm $R \sim 3\,R_\oplus$ planet orbiting α Centauri A. For comparison, the plausible radii range of rocky exoplanets extends up to $R \sim 1.75\,R_\oplus$[35,36].

## Discussion

The primary goal of the NEAR campaign is to demonstrate the capabilities of mid-IR exoplanet imaging. The results showed that the sensitivity is background limited and follows a signal to noise ratio (SNR) $\propto \sqrt{t}$ relation in image regions far from the center ($\geq 7\,\lambda/D$). The achieved sensitivity in such regions in one hour of observations (5σ) is $\sim 0.75$ mJy (see ref. [37], and supplementary methods, background-limited sensitivity). The habitable zone of α Centauri A is located at $\sim 1''$ (see Fig. 1 and ref. [13]), which corresponds to the contrast-limited region of $\sim 3.5\,\lambda/D$ in the 10–12.5 μm bandpass for the 8.2-m VLT. With a 39-m Extremely Large Telescope (ELT), $\sim 1''$ would correspond to $\sim 17.5\,\lambda/D$ and would therefore likely be close to background-limited. In that case, the SNR scales $\propto \sqrt{t}D^2$, and thus the time required to reach a given SNR scales $\propto D^{-4}$. The predicted sensitivity at $1''$ of a

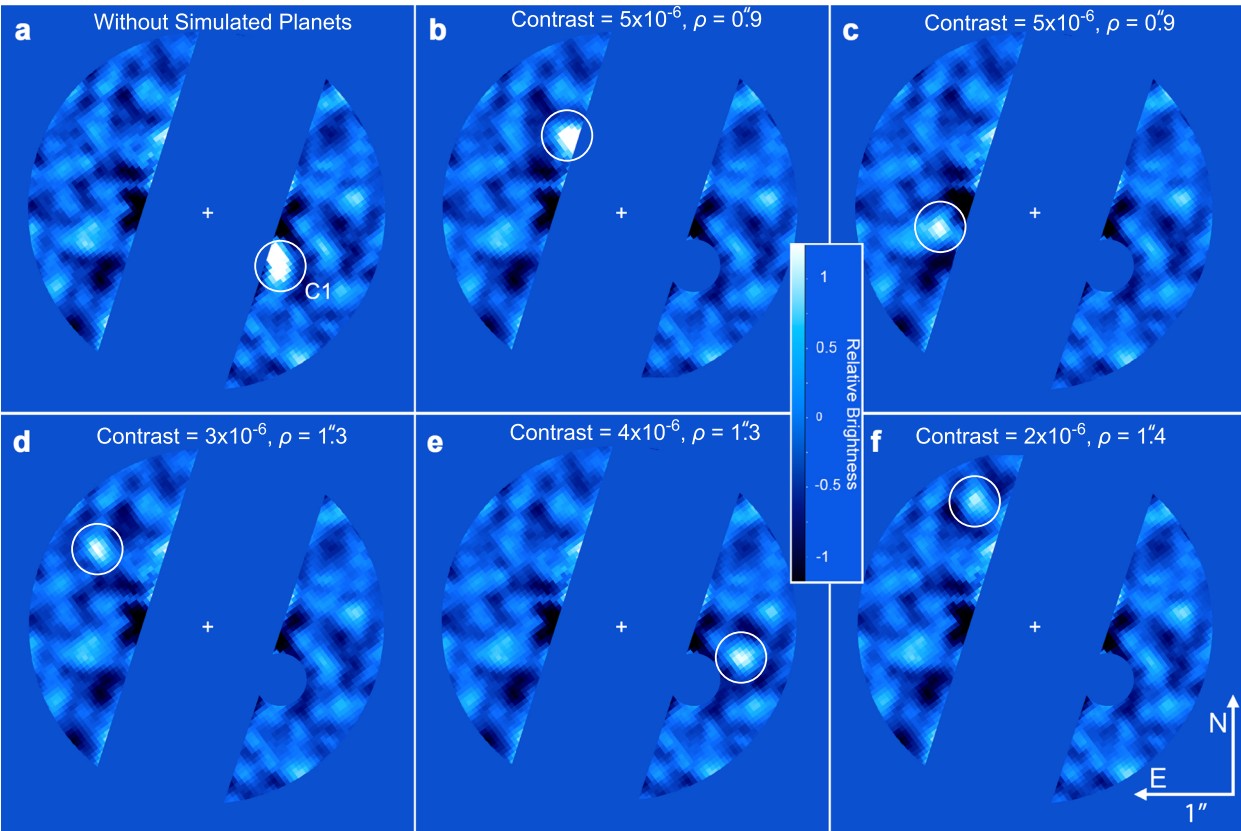

**Fig. 3 Simulated planet injections.** NEAR campaign image (**a**) and those with simulated planets (**b–f**). Each image has been PSF-subtracted following removal of known artifacts. The location of C1 has been masked in (**b–f**) so that the simulated planets (indicated in these panels by white circles) can be clearly identified. These examples demonstrate the lower brightness limit at which simulated planets are identifiable. The bottom right panel represents the limiting case at which the source is marginally identifiable among speckles of similar brightness.

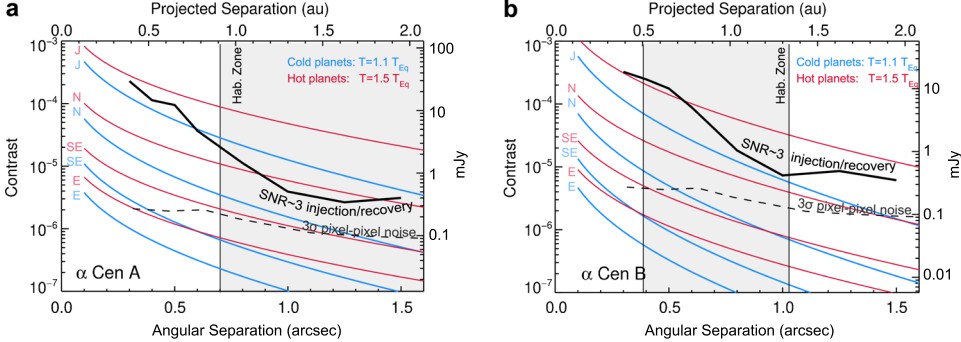

**Fig. 4 Sensitivity of the NEAR data.** Results for α Centauri A & B are shown in (**a**) and (**b**), respectively. The bold curve shows the sensitivity to point sources computed from simulated injection and recovery tests and the dashed curve shows the background noise contribution from the variation of pixel-to-pixel intensities. The red and blue curves represent simulated planets of equivalent radii to those in the solar system, with the addition of an $R = 1.7\,R_\oplus$ Super-Earth (SE). Each model planet's temperature is set by the assumption of thermal equilibrium at a given separation with an $A_B = 0.3$ Bond albedo and internal heating included as 10% or 50% of the equilibrium temperature; similar to conditions of the solar system planets[46,47].

NEAR-like instrument on an ELT would therefore be ~35 μJy (5σ in 1 h). This would in principle be sufficient to detect an Earth-analog planet around α Centauri A (~20 μJy) in just a few hours, which is consistent with expectations for the ELTs[7,38].

If the ELT's performance at 1″ is instead contrast-limited, then Earth-like planets could still be imaged, since the intensity of quasi-static speckles produced by optical polishing errors at a given angular separation scales $\propto D^{-2}$ or steeper[39]. The NEAR campaign demonstrated a final contrast-limited sensitivity (SNR~3 in 77 h) of ~3 × 10⁻⁶ contrast to α Centauri A (~0.4 mJy) at ~1″. Extrapolating to the larger aperture of the ELT suggests a contrast limit of ~1.5 × 10⁻⁷ or better at 1″. This again supports the predictions that the ELTs will reach sensitivity levels sufficient to image Earth-analog planets around α Centauri A[7,38]. These estimates may also be expected to be improved, since the increased local background produced near the center due to the glow of the AGPM can be mitigated by a cold pupil stop in front of the coronagraphic mask, as implemented by the current instrument design plans for the METIS instrument[40]. The contrast-limited performance of future instruments could also be improved by pupil apodizers[41] and non-common path aberration calibration mechanisms[42] that were not available for the design of NEAR.

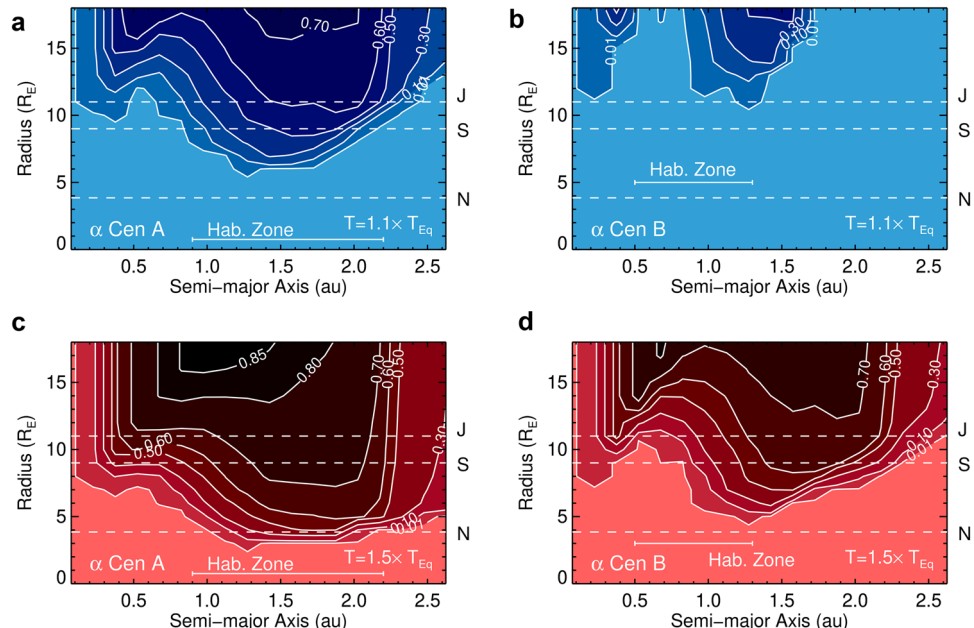

**Fig. 5 Completeness to planets of various radii and orbital semi-major axes.** (**a**) and (**b**) assume $A_B = 0.3$ and internal heating contributing 10% of the equilibrium temperature, whereas (**c**) and (**d**) assume $A_B = 0.3$ and internal heating contributing 50% of the equilibrium temperature. Radius and semi-major axis were uniformly sampled along with an inclination prior of $P(i) \propto \sin i$ (see supplementary methods, completeness analysis). The dashed lines correspond to the radii of Neptune, Saturn, and Jupiter (N, S, and J, respectively). Contour units are normalized.

Improvements to mid-IR detector technologies could also lead to significant improvements.

A secondary goal of NEAR is to explore the habitable zones of α Centauri. While designed to detect thermal emission from exoplanets, our observations could also detect warm exozodiacal dust (e.g., refs. [43]). Here, we consider whether either such detection is present in the images. In a relatively clean region of the image, there is one point-like feature (SNR ∼ 3) that is not associated with any known detector artifacts. We refer to this source as Candidate 1, or C1 (see Fig. 2). C1 appears with a brightness that would be expected of a giant planet ($R \sim 3–11 \, R_\oplus$) at ∼1.1 au from α Centauri A and with an elongation on the order of ∼0.1 arcsec–consistent with orbital motion of a planet in an $i \sim 70°$ orbit throughout the nineteen-day campaign. Notably, the detection of C1 is repeatable in multiple independent subsets of the data (see Supplementary Fig. 2), which makes it unlikely to be a random false positive. Based on pre-imaging conducted a decade prior to the NEAR campaign, we can exclude the possibility that C1 is a background source (see ref. [44] and supplementary methods, pre-imaging for background sources). Therefore, we consider C1 to be a plausible exoplanet and/or exozodiacal disk candidate. While C1 cannot be explained by presently known systematic artifacts, an independent experiment is necessary to exclude this third possibility.

RV observations exclude the presence of $M\sin i \geq 53 \, M_\oplus$ planets within the habitable zone α Centauri A[14]. Assuming $R \propto M^{0.55}$[45], this limit corresponds to $R < 7 \, R_\oplus$. Among a range of radii of $R \sim 3.3–7 \, R_\oplus$, the brightness of C1 can be explained with a level of additional heating sufficient to raise the planet's temperature by 5–50% of its radiative equilibrium temperature (assuming $A = 0.3$). The lower limit is motivated by Neptune's effective temperature, which is ∼50% higher than its radiative equilibrium temperature[46,47]. C1 could also be an exozodiacal disk with ∼60 zodis of dust, and with a stellocentric offset of ∼0.3 au to the SW (see supplementary methods, exozodiacal dust disk modeling). This would be a relatively large dust mass for a G-type star[48], but would be within precedent (e.g., ε Eridani has ∼200 zodis:[48]). This dust mass is also consistent with the upper limits from the far-IR

spectrum of α Centauri A ($\leq 100$ zodis,[49]). In other words, C1 is not a known systematic artifact, and is consistent with being either a Neptune-to-Saturn-sized planet or an exozodiacal dust disk.

The habitable zones of α Centauri and other nearby stars could host multiple rocky planets–some of which may host suitable conditions for life. With a factor of two improvement in radius sensitivity (or a factor of four in brightness), habitable-zone super-Earths could be directly imaged within α Centauri. An independent experiment (e.g., a second mid-infrared imaging campaign, as well as RV, astrometry, or reflected light observations) could also clarify the nature of C1 as an exoplanet, exozodiacal disk, or instrumental artifact. If confirmed as a planet or disk, C1 would have implications for the presence of other habitable zone planets. Mid-infrared imaging of the habitable zones of other nearby stars, such as ε Eridani, ε Indi, and τ Ceti is also possible. In the next decade, the application of these techniques with extremely large telescopes (e.g., with ELT/METIS:[7,38,40]) will enable sensitive exploration of the habitable zones of these and other nearby stars.

## Methods

**Instrumental setup and observing strategy.** The VLT Imager and Spectrometer for the Mid-IR (VISIR:[25]) was significantly upgraded for the NEAR experiment. VISIR was coupled with the VLT's DSM[31], which enabled the implementation of AO without increasing the number of warm optics that would add to the thermal background. The AO correction resulted in typical Strehl ratios in excess of 97%. The DSM was used for ∼8 Hz chopping to enable tracking and subtracting the systematic excess low-frequency noise (ELFN) within the Si:As Aquarius detector, which is a major limitation to the sensitivity of mid-infrared imaging[50,51]. Downstream of the DSM, the central starlight was reduced by an AGPM coronagraph optimized for performance at mid-infrared wavelengths[26,27] and a shaped-pupil mask[28–30] designed specifically to limit the spatial extent of the Airy pattern from the off-axis star. The Lyot stop is manufactured out of chromium directly deposited on the NEAR spectral filter, which transmits light from 10 to 12.5 μm. This yields a full width at half maximum (FWHM) of ∼0.28 arcsec, or ∼6 pixels. The observations were done in a pupil-stabilized mode by keeping the Cassegrain instrument at a fixed rotation angle. The detector integration time (DIT) was 6 ms, of which eight frames were averaged and two frames were skipped during the chopping transition. Therefore, each chopping half-cycle equated to 60 ms, resulting in a chopping frequency of 8.33 Hz. For the night of 2019 May 24 we used a DIT of 5.5 ms and normalized the images for this night to account for the difference.

**Data Reduction and Processing**. We reduced the data for each night of the campaign in a uniform manner with two independent pipelines, which we refer to as the primary and secondary data reductions. The secondary reduction does not implement artifact modeling and subtraction. Therefore, in most cases we utilize the higher fidelity images from the primary reduction and utilize the secondary reduction to confirm the general findings of the first.

We begin by describing the primary data reduction pipeline. From each individual frame with α Centauri A behind the coronagraph we subtracted the mean of the two neighboring frames (chop subtraction). No scaling was performed to normalize the PSFs, as the purpose of chop subtraction is primarily to remove the ELFN and residual background structure such as the AGPM glow. The residual coronagraphic PSF is also partially mitigated by chop subtraction. We then coadded each five hundred image cube into a single image with 24 s of equivalent exposure time and combined each of these frames into a single data cube per night of observations. We aligned the frames within each cube via the unocculted PSF of α Centauri A, and determined the precise center of the coronagraphic residual of A-B via rotational centering[52]. We cleaned the frames by rejecting those whose maximum cross-correlation with respect to the mean of the twenty surrounding frames was less than 0.9 (computed over the radial range of 5–45 pixels, or ∼0.2–2 arcsec from the center), which resulted in ∼10% frame rejection. At this stage, the known detector artifacts were subtracted from the images (see supplementary methods, artifact modeling I and II). We destriped the images along the horizontal and vertical axes by subtracting the mode of each row and column, and high-pass filtered the data by subtracting a version of each frame from itself after smoothing with a 15-pixel running median. We then stacked and averaged the original frames into 360 s images, and processed the data via both classical angular differential imaging (ADI:[53]) and projection onto eigen images via Karhunen–Loève Image Processing (KLIP: [54]; specifically using the adaptation from[55]), in which we modeled the PSF with four KL-modes in an annulus from 5–45 pixels. One beam diameter corresponds to ∼14° in azimuth at 1 arcsec, which is significantly larger than the 2.2° of smearing introduced in 360 s due to the rotation of the sky. Following PSF subtraction, we applied a second high-pass filter with the same settings to reduce the remaining low-frequency spatial variations. We combined the images within each night using a noise-weighted combination for each pixel, (noise-weighed ADI:[56]) and combined the final images from each night with a variance-weighted mean. The images are shown in Figs. 2, 3 and Supplementary Figs. 2, 3.

For the second data reduction pipeline, we followed a similar procedure with the following exceptions. Various quality criteria were calculated for the individual chopped images, including AO correction (ratio of flux in an annulus of radii 6–12 pixels to an aperture of 6 pixels radius), coronagraphic leakage (flux in an aperture of 20 pixels) and sky-background variance calculated over small regions near the edge of the frame. 79.3 h of data remained after removing the inferior images, which is a similar total exposure time as for the data set created by our other pipeline. Then, the images were co-aligned to the center between the off-axis positions of α Cen A and B and then mean-combined to create frames with an equivalent exposure time of 60 s. We then calculated an ADI-based principal component analysis (PCA) model (e.g., ref. [57]) over an annular region around the image center and used this model to subtract the PSF for each observing night separately. Using fake planet injection tests, we optimized the PCA parameters (inner and outer radius of the annulus, number of principal components) to maximize the contrast sensitivity. We arrived at using 15 principal components, an inner radius of 8 px, and an outer radius of 16 px (although the regions further out are also processed). We verified that our conclusions are robust over a range of a few principal components to the maximum number of frames. The final image quality is quite robust with respect to the selected optimization range, and variations of 50% of each parameter do not significantly affect the results. Finally, the images for each night were combined with a variance-weighted mean (Supplementary Fig. 3). Before subtraction of artifacts, both pipelines deliver comparable performance.

## Data availability

All data from the NEAR campaign are publicly available at archive.eso.org under program ID 2102.C-5011(A). Original and processed data are also available from the corresponding author upon request.

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

## Acknowledgements
NEAR was made possible by contributions from the Breakthrough Watch program, as well as contributions from the European Southern Observatory, including director's discretionary time. Breakthrough Watch is managed by the Breakthrough Initiatives, sponsored by the Breakthrough Prize Foundation. We would like to thank Rus Belikov, Eduardo Bendek, Bernhard Brandl, Ryan Endsley, Rachel Fernandes, Kaitlin Kratter, Christian Marois, Michael Meyer, and Maxwell Moe for fruitful discussions and advice. The results reported herein benefited from collaborations and/or information exchange within NASA's Nexus for Exoplanet System Science (NExSS) research coordination network sponsored by NASA's Science Mission Directorate. This work was supported by the National Centre of Competence in Research PlanetS supported by the Swiss National Science Foundation, by the Fonds de la Recherche Scientifique—FNRS under Grant no F.4504.18, by the European Research Council (ERC) under the European Union's Horizon 2020 research and innovation program (grant agreement no. 819155) and under the European Union's Seventh Framework Program (grant agreement no 337569), and by the Wallonia-Brussels Federation (grant for Concerted Research Actions). K.W. acknowledges support from NASA through the NASA Hubble Fellowship grant HST-HF2-51472.001-A awarded by the Space Telescope Science Institute, which is operated by the Association of Universities for Research in Astronomy, Incorporated, under NASA contract NAS5-26555. A.B. and S.P.Q. acknowledge the financial support of the Swiss National Science Foundation. G.R. was supported by an NSF Astronomy and Astrophysics Postdoctoral Fellowship under award AST-1602444.

## Author contributions
M. Kasper contributed as the NEAR experiment lead. K.W., A.B., P.P., M. Kasper, É.C., and S.P.Q. contributed to the data analysis. K.W., A.B., P.P., M. Kasper, and D.A. contributed to the preparation of the manuscript. K.W., A.B., U.K., A.-L.M., E.P., R.S., O.A., N.A., D.A., O.G., D.M., S.P.Q., M.S., A.V., and T.Z. contributed scientific advice. M. Kasper, U.K., A.-L.M, E.P., R.S., G.Z., O.A., and J.K. contributed to campaign observations. R.A. contributed to experiment management. G.J. contributed to experiment engineering (mechanics, system). U.K. contributed to experiment engineering (optics, system). A.-L.M., O.A., P.F., E.H., M. Karlsson, D.M., G.O.X., and G.R. contributed to experiment engineering (coronagraph). A.C. and G.R. contributed to coronagraph apodizer design. G.Z. and P.D. contributed to experiment engineering (software). N.A. N.A., S.G., D.K., S.L., A.R., and M.R. contributed to experiment engineering (system). C.D. contributed to experiment testing (coronagraph). K.D., M.N.D., and A.V. contributed to experiment engineering (non-common path aberrations). E.F. contributed to experiment engineering (cooling). O.G. and P. Klupar contributed to experiment oversight. P. Kervella contributed to the analysis of the background star hypothesis. J.-P.K. contributed to experiment engineering (mechanics). C.S. contributed to experiment engineering (electronics). J.K. contributed to adaptive optics operation. M.N.D. and M.R. contributed to experiment commissioning. M.S. and T.Z. contributed to project oversight.

## Competing interests
The authors declare no competing interests.
