## [Peer Review File · Nature Communications]

REVIEWER COMMENTS

Reviewer #1 (Remarks to the Author):

The paper describes an impressive 100 hour campaign designed to search for exoplanets around Alpha Cen A and B at 10 microns. This is the first serious attempt to image a low-mass exoplanet by using current technologies and facilities, pushed to the practical limits of integration time. A candidate is discovered, which appears to be astrophysical. However, assuming it is real, it is too bright to be a rocky planet. This is clearly an interesting outcome of the paper, but it will need follow-up study to determine whether it is an exozodiacal disk, an exoplanet, or something else.

While the study does not reach the sensitivities necessary to detect a habitable planet, it successfully demonstrates many of the technologies that will be necessary to image a habitable exoplanet with the next generation of extremely large telescopes, such as the ELT. In this regard, I feel that the paper would benefit from an expanded discussion of the ramifications of the NEAR campaign for analogous observations with ELTs. What were the limiting noise sources? Can they be mitigated or are they intrinsic to the technique or technologies? Given these mitigations, what are the prospects for imaging rocky planets with the ELTs? I'll note that looking at Figure 4, and assuming a $\sim 20\times$ improvement in S/N going from a VLT aperture to an ELT aperture, it would be just barely possible to image an Earth-sized planet around Alpha Cen A at 3-sigma with $T=1.5\times T_{\text{eq}}$ (which is quite optimistic) and it would not be possible to image a $T=1.1\times T_{\text{eq}}$ planet around Alpha Cen A, or either type of planet around Alpha Cen B. In other words, while 10-micron imaging of Earth-like planets is a stated goal of the ELTs, the NEAR experiment did not demonstrate the sensitivity necessary to make such observations practical for the ELTs, unless there are further technological improvements.

Overall, I think this paper is a thorough summary of an important scientific campaign to push the limits of exoplanet imaging below gas-giant exoplanets. Since the current campaign is not sensitive enough to detect rocky planets, part of the scientific importance of this work will be to motivate analogous studies with the ELTs. It is widely discussed that, if limited by sky-background Poisson noise, the ELTs will have the sensitivity and angular resolution to see Earth-like planets around a handful of nearby stars. The NEAR survey takes a step in this direction, but ultimately, understanding and mitigating the current experiment's systematics will be important.

I have some smaller comments listed below. Overall, I was quite impressed by the detail and tests described in the Supplement section. For the most part, my comments refer to the Main Text, which I read (and wrote my comments for) first. Some of my questions and comments there are addressed in the Supplement, and the authors may choose to refer readers to the Supplement in those cases.

-Andy Skemer

PAGE 2

Line 24—Give wavelength range for “optical to near-infrared wavelengths”

Line 26—Give wavelength range for “mid-infrared”

Line 33—Given the broad audience for this paper, you might say that the coarser spatial resolution is due to diffraction.

PAGE 3

Paragraph starting on Line 6—Alpha Cen C (aka Proxima Cen) does have exoplanets and should be mentioned here.

Figure 1, Left Panel—Is this really the Projected Separation? Or is it Physical Separation? If the latter, it would be more clear to label the x-axis “Face-On Separation (arcsec)”

Line 32—You should explain the non-cryogenic optics contribute to the backgrounds emission and decrease sensitivity.

PAGE 5

Line 9—Pixel-to-pixel noise variations would not ordinarily be the correct metric for measuring the sensitivity to planets, since the PSF covers multiple pixels. If noise is correlated on PSF-sized scales, then this might still give you the right answer, but it would be more straight-forward to calculate sensitivity on PSF-sized apertures, rather than pixels.

Line 13—Your sensitivity to point-sources would not be similar to the 1-sigma noise level (even in the purely background limited case, with no residual structures), because each image contains ~100 apertures (I’m guessing). Therefore, you would need at least ~4-sigma (1/15,000 false positives/negatives) if you had perfectly Gaussian errors, which would give you 1/150 chance of detecting a false positive/negative in an image with 100 apertures. You probably should go through an argument like this to define your detection threshold (you show 3-sigma in Figure 4, which may be OK, but you would expect false positives frequently)

PAGE 6

Figure 4, last sentence—needs citation

Figure 5—When you say “no prior constraints were assumed for semi-major axis”, I assume the figure still encapsulates the unknown inclination angle of the system and the fact that a planet on an inclined orbit would be harder to detect? It looks like your text says that, but you might want to add a sentence to the caption.

Figure 5—Related to the previous point, you should assume some sort of a prior on inclination. Either something related to the orbit of the binary, or at a minimum, $P(i)$ is proportional to $\sin(i)$ for random sky orientations.

PAGE 7

Line 23—give the radius (or range of radii) of the putative giant planet.

SUPPLEMENT:

When doing background subtraction, how do you compensate for Alpha Cen A and B being different brightnesses?

If the planetary systems are aligned with the binary orbit then the masked region of Figure 2 is precisely where you would expect to detect planets. Luckily this does not appear to be the case. You may want to reference some of the recent ALMA work that discussed the co-alignment of binary systems.

Reviewer #2 (Remarks to the Author):

The authors present a novel observational effort to directly image low-mass planets in the nearest stellar system, alpha Centauri. The team has invested an unprecedented amount of time at the Very Large Telescope for these efforts—77 hours of on-source integration time over the course of several weeks—which is nearly two order of magnitude longer than the typical integration time used in traditional high-contrast imaging surveys. The observations were carried out at mid-infrared wavelengths (10 microns) where thermal emission from low-mass planets is highest. This is challenging because Earth’s atmosphere and telescope optical surfaces are also bright in this region, requiring unique technical setup and considerations. Moreover, alpha Cen is a modest-separation binary (with a wide M dwarf tertiary), so a creative observing and processing strategy was needed.

As a feasibility study to explore new parameter space (both in general and around these stars), this seems to have been a success. With reasonable assumptions about planet albedo, thermal evolution, and internal heating, this experiment appears to have reached sub-Saturn sensitivities throughout much of the classically-defined habitable zone of alpha Cen, and down to Neptune sized planets in the most sensitive region—the best mass sensitivity limits with direct imaging for any system, as far as I am aware. This is impressive and in many ways transformational compared to the standard approach to direct imaging surveys over the past two decades, which has been limited to Jovian-mass planets around young stars. This also serves as a preview of future possibilities in the mid-infrared with the upcoming thirty-meter class telescopes and, soon, JWST.

My primary concern with this study relates to the interpretation of the emission feature located near the inner edge of the habitable zone and which the authors attribute to a potential planet or warm circumstellar dust. The authors have carried out a series of experiments to gauge the significance and robustness of this feature. They arrive at a SNR of 3.1-3.5. The residuals after PSF subtraction in high-contrast imaging datasets are plagued by non-Gaussian noise statistics as well as systematics from imperfect AO correction, so any detection with a SNR less than ~ 5 should be treated with extreme caution. This is mitigated at some level by the long wavelengths and high Strehl ratios in this study (which should be quoted). The signal also appears to be present after splitting the campaign data in half, and after processing even-odd nights (although no significance level is given for these, as far as I can tell, but presumably they have $\text{SNR} < 3$). However, there are many sources of systematics like persistence stripes and ghosts in this dataset. Other apparent sources appear to be present in the processed images near the circular ring marking the habitable zone inner edge but before the wide strip covering the persistence stripes is masked out. These are evident in Figure S1 (B, E, and F panels) and Figure S3 (A and B panels) at ~ 11 o'clock and ~ 2 o'clock. Given the low SNR of the detection and the other artifacts in the image, it is not clear to me that the "C1" feature is any different.

Other minor comments:

(1) The integration time of the data that were used was 77 hours, not 100 hours. 100 hours should be removed from the abstract.

(2) Please include how habitable zone is being defined, as this can dramatically vary depending on assumptions about a planet's atmospheric pressure and greenhouse effects.

(3) There is a substantial difference (by ~ 2 orders of magnitude!) between the 3 sigma pixel noise contrast curve and the 3 sigma injection-recovery contrast curve in Figure 4, especially at small

angular separations. The injection-recovery approach should be more robust as it mimics the practice of actually detecting point sources. Because of the large discrepancy between the two approaches, the pixel noise curve should not be displayed in this figure.

(4) Please explain where the Bond albedo of 0.3 comes from (presumably Earth).

Reviewer #3 (Remarks to the Author):

This is an intriguing paper. It presents some novel aspects of IR coronagraphic imaging of alpha Cen A and B and reports a possible object in orbit around A. The work is excellent, but lacks some important aspects in the analysis of the candidate they call C1.

The Technique

The authors present a novel way of using chopping in 10 micron, ground based imaging by exploiting a chopping secondary (actually a deformable mirror). The technique is sound and in principle could be replicated elsewhere (with significant investment by other observatories).

The other novelty is using the fact that A and B are only 15 arcseconds apart, so they partially serve as PSF reference stars for each other in the image data reduction scheme. This is quite clever, and probably can be improved upon with some other tricks, and can be used in other attempts to image substellar objects in binary systems, an area that is ripe for discoveries, but has been technically difficult, although methods are being studied by others as well.

The authors argue that with their coronagraph and the PSF removal, followed by some artifact removal and pesky detector persistence, they would be able to image planets as small as Saturn, or maybe even super-earth sized ones around these two stars, if they had roughly 100 hours of observing time. This was not entirely convincing.

First, the sensitivity analysis uses a very crude model for the brightness of these simulated planets, which is basically a black body function plus some chosen albedo, although, even with the

supplemental material, the model is not reproducible by a reader. They try several variations on the internal heating of their putative planets. However, we know for a fact that Earth, Jupiter, Neptune etc are not black bodies and in fact at these wavelengths there are many (quite interesting, actually) molecular absorption features that deviate the spectral energy distribution of the emergent spectra considerably from a black body function. The main issue here is that this is the fundamental basis for their argument that given 100 hours of VLT time to do this experiment again, they could detect such things. Everything hinges on the model flux density in the N-band ($\sim 10\mu\text{m}$). The description, even in the supplemental material, is decidedly vague and does not even give what temperatures were used for each object or what radii were used, what irradiance they used, etc. Regardless, the model is so simplistic that it evokes significant doubt in the results. In addition, simply amping up internal heat sources would have a significant effect on the atmospheres of these things as well as their emergent spectra.

The Candidate Companion

It is extremely exciting that the team found a candidate companion to Alpha Cen A, which they call C1. Whatever it is, it is an important discovery, if it is orbiting A. It does not need to be a Jupiter radius planet to be exciting. For example, brown dwarfs are also about that radius, and finding one in this system would be equally important.

However, although the detection of something next to Alpha Cen A seems certain, the arguments that it is orbiting A were again too simplistic. A simple quick calculation suggests that its small motion from May to June is actually quite consistent with the motion of A on the sky, although I am confused by the images presented. A is about ~ 15 arcsec to the NE of B, so it seems that the compass on the image figure may be mistaking the cardinal directions (is N really down, not up as indicated?). It would be nice to label which residuals belong to which star when it is unocculted in Fig. 2A (as done in the reference 25 paper), and it would be helpful if figure 1B were actually oriented the way the system is on the sky. (N up, E to the left, is the convention, although of course this was done in the southern hemisphere.)

Even if the orientation of the images is correct, it is important to consider the motion of A over the period of observations. Three different dates of observations were separated by 2.5 and 2 weeks respectively (roughly). A has a proper motion of ~ 170 mas South and ~ 33 mas E in 2.5 weeks ($-3608, 686$)mas/yr. This is actually consistent with the delta position mentioned in the supplementary material (if those directions are correct). However, on top of this, A also has a parallax of 742 mas. In 2.5 weeks, the parallactic motion then is $2 \times 35.7 \text{ mas} = 71$ mas in two weeks (actually more, since it is so far south and its parallactic motion will be nearly circular, but this is just a rough estimate). Whether these cancel each other out or actually magnify each other, the authors need to consider these issues, and a plot of the motion on the sky would be helpful. There is also the third component which is the motion around the AB barycenter. It is possible that doing this analysis properly makes the case for C1 to be a candidate even much stronger. The best and only certain method of

companion confirmation is common parallax, which requires gravitational interaction. So at some level, the authors may have a more secure result than they know, although this is complex and has to be analyzed properly, including all of the motions involved.

I did not find the pre-imaging argument using K band data in 2009 convincing. It could be with some figures, perhaps, showing how A is moving around over that period of time (i.e. not just proper motion).

Finally, there are many examples of claims of finding companions that later turned out to be irreproducible or incorrect, something I am sure the authors are aware of. It is a bit of a warning sign that this thing was not detected in the third data set, although certainly the argument that it moved into the bad part of the field of view is possibly true. It would be wise for the authors to obtain at least a few more similar data sets before arguing (as they do) that this C1 must be in orbit around A.

General Comments:

I found the paper to be written stylistically like a proposal to the VLT Time Allocation Committee to get 100 hours of telescope time. That is fine, but the authors should be aware of that impression. However, I hope these comments in this review help the authors to make this a much stronger paper. It was certainly exciting to read, but it needs more work.

REVIEWER COMMENTS

Reviewer #1 (Remarks to the Author):

The paper describes an impressive 100 hour campaign designed to search for exoplanets around Alpha Cen A and B at 10 microns. This is the first serious attempt to image a low-mass exoplanet by using current technologies and facilities, pushed to the practical limits of integration time. A candidate is discovered, which appears to be astrophysical. However, assuming it is real, it is too bright to be a rocky planet. This is clearly an interesting outcome of the paper, but it will need follow-up study to determine whether it is an exozodiacal disk, an exoplanet, or something else.

We agree that the candidate requires follow-up to determine its true nature. We also agree that it appears to be astrophysical, but that we cannot exclude other possibilities. This much was already addressed in the paper, as the reviewer clearly understood. Unfortunately, follow-up is not possible at the moment, although several potential avenues are being explored (see our responses to related comments below).

While the study does not reach the sensitivities necessary to detect a habitable planet, it successfully demonstrates many of the technologies that will be necessary to image a habitable exoplanet with the next generation of extremely large telescopes, such as the ELT. In this regard, I feel that the paper would benefit from an expanded discussion of the ramifications of the NEAR campaign for analogous observations with ELTs. What were the limiting noise sources? Can they be mitigated or are they intrinsic to the technique or technologies? Given these mitigations, what are the prospects for imaging rocky planets with the ELTs? I'll note that looking at Figure 4, and assuming a $\sim 20\times$ improvement in S/N going from a VLT aperture to an ELT aperture, it would be just barely possible to image an Earth-sized planet around Alpha Cen A at 3-sigma with $T=1.5\times T_{\text{eq}}$ (which is quite optimistic) and it would not be possible to image a $T=1.1\times T_{\text{eq}}$ planet around Alpha Cen A, or either type of planet around Alpha Cen B. In other words, while 10-micron imaging of Earth-like planets is a stated goal of the ELTs, the NEAR experiment did not demonstrate the sensitivity necessary to make such observations practical for the ELTs, unless there are further technological improvements.

This is an excellent suggestion. Since the ELTs will be background-limited at $1''$, the situation is actually more optimistic than a simple scaling of the VLT's contrast limit at $1''$. We have added the following expanded discussion on the ramifications of the NEAR experiment for thermal infrared exoplanet imaging with the ELTs:

“The primary goal of the NEAR campaign is to demonstrate the capabilities of thermal infrared exoplanet imaging. The results showed that the sensitivity is background limited and follows a $\propto \sqrt{t}$ relation in image regions far from the center ($\geq 7 \lambda/D$). The achieved sensitivity in such regions in one hour of observations (5σ) is ~ 0.75 mJy (see 36, and Supplementary Methods). The habitable zone of α Centauri A is located at $\sim 1''$ (see Figure 1 and 19), which corresponds to the contrast-limited region of $\sim 3.5 \lambda/D$ in the 10–12.5 μm bandpass for the 8.2-m VLT. With a 39-m Extremely Large Telescope (ELT), $\sim 1''$ would correspond to $\sim 17.5 \lambda/D$

and would therefore likely be close to background-limited. In that case, the SNR scales $\propto \sqrt{t} \cdot D^2$, and thus the time required to reach a given SNR scales $\propto D^{-4}$. The predicted sensitivity at 1" of a NEAR-like instrument on an ELT would therefore be $\sim 35 \mu\text{Jy}$ (5σ in one hour). This would in principle be sufficient to detect an Earth-analogue planet around α Centauri A ($\sim 20 \mu\text{Jy}$) in just a few hours, which is consistent with expectations for the ELTs (7, 37).

If the ELT's performance at 1" is instead contrast-limited, then Earth-like planets could still be imaged, since the intensity of quasi-static speckles from optical polishing errors at a given angular separation scales $\propto D^{-2}$ or steeper (38). The NEAR campaign demonstrated a final contrast-limited sensitivity (SNR ~ 3 in 77 hours) of $\sim 3 \times 10^{-6}$ contrast to α Centauri A ($\sim 0.4 \text{ mJy}$) at $\sim 1''$. Extrapolating the contrast limit to the larger aperture of the ELT would suggest a contrast limit of $\sim 1.5 \times 10^{-7}$ or better at 1". This again supports the predictions that the ELTs will reach sensitivity levels sufficient to image Earth-analogue planets around α Centauri A (7, 37). These estimates may also be expected to be improved, since the increased local background produced near the center due to the glow of the AGPM can be mitigated by a cold pupil stop in front of the coronagraphic mask, as implemented by the current instrument design plans for the METIS instrument (39). The contrast-limited performance of future instruments could also be improved by pupil apodizers (40) and non-common path aberration calibration mechanisms (41) that were not available for the design of NEAR. Finally, improvements to mid-IR detector technologies could also lead to significant improvements."

Overall, I think this paper is a thorough summary of an important scientific campaign to push the limits of exoplanet imaging below gas-giant exoplanets. Since the current campaign is not sensitive enough to detect rocky planets, part of the scientific importance of this work will be to motivate analogous studies with the ELTs. It is widely discussed that, if limited by sky-background Poisson noise, the ELTs will have the sensitivity and angular resolution to see Earth-like planets around a handful of nearby stars. The NEAR survey takes a step in this direction, but ultimately, understanding and mitigating the current experiment's systematics will be important.

We agree with the reviewer that part of this work's importance is to motivate future work with the ELTs and to explore their potential scientific capabilities. We have taken these comments into consideration with the expanded section on ELTs copied above.

I have some smaller comments listed below. Overall, I was quite impressed by the detail and tests described in the Supplement section. For the most part, my comments refer to the Main Text, which I read (and wrote my comments for) first. Some of my questions and comments there are addressed in the Supplement, and the authors may choose to refer readers to the Supplement in those cases.

-Andy Skemer

We've updated this to “wavelengths of $\lambda \leq 5 \mu\text{m}$.”

Line 26—Give wavelength range for “mid-infrared”

We've added “($\lambda \sim 10\text{--}20 \mu\text{m}$).”

Line 33—Given the broad audience for this paper, you might say that the coarser spatial resolution is due to diffraction.

We've added “due to the diffraction limit scaling with wavelength”

PAGE 3

Paragraph starting on Line 6—Alpha Cen C (aka Proxima Cen) does have exoplanets and should be mentioned here.

We've added the following:

“The tertiary M-dwarf component of the system, Proxima Centauri, hosts at least two planets more massive than Earth (58, 59) that were discovered through the star's radial velocity (RV) variations.”

Figure 1, Left Panel—Is this really the Projected Separation? Or is it Physical Separation? If the latter, it would be more clear to label the x-axis “Face-On Separation (arcsec)”

The reviewer is correct that the top axis should be labeled “Physical Separation”. This change has been made and we have also renamed the bottom axis to “Face-on Angular Separation (arcsec)”. We have also indicated in the caption that the simulated planet orbits are face-on and circular.

Line 32—You should explain the non-cryogenic optics contribute to the backgrounds emission and decrease sensitivity.

We've added: “without additional non-cryogenic corrective optics whose thermal emission would contribute to the total background.”

PAGE 5

Line 9—Pixel-to-pixel noise variations would not ordinarily be the correct metric for measuring the sensitivity to planets, since the PSF covers multiple pixels. If noise is correlated on PSF-sized scales, then this might still give you the right answer, but it would be more straight-forward to calculate sensitivity on PSF-sized apertures, rather than pixels.

The reviewer is correct that the pixel-to-pixel noise variations are not the correct metric for assessing the sensitivity to point sources. Our intention was to assess and report the pixel-to-pixel noise level as a fundamental sensitivity limit in the absence of other sources of noise. We have clarified this by rewriting the ending of this paragraph in the following

manner:

“As an assessment of the detector’s fundamental sensitivity limit (in the absence of residual stellar flux and spatially correlated noise), we examined the standard deviation of pixel intensities within $1.2 \lambda/D$ (~0.35 arcsec, or ~8 pixels) in diameter in a region of the detector far from α Centauri A and B (see SOM). We found this value (multiplied by the square root of the number of pixels contained within the aperture) to be $\sim 1.67 \times 10^{-7}$ contrast with respect to α Centauri A, or about $\sim 22 \mu\text{Jy}$. The pixel-to-pixel noise increases toward the glow of the AGPM (29). At 1" separation, the standard deviation of pixel intensities is roughly doubled by the glow.”

Line 13—Your sensitivity to point-sources would not be similar to the 1-sigma noise level (even in the purely background limited case, with no residual structures), because each image contains ~100 apertures (I’m guessing). Therefore, you would need at least ~4-sigma (1/15,000 false positives/negatives) if you had perfectly Gaussian errors, which would give you 1/150 chance of detecting a false positive/negative in an image with 100 apertures. You probably should go through an argument like this to define your detection threshold (you show 3-sigma in Figure 4, which may be OK, but you would expect false positives frequently)

The referee is correct about this point. We have removed this sentence and references to Gaussian-distributed uncertainties. The residuals in the vicinity of the image center are clearly non-Gaussian given the level of systematic structures (most notably the persistence stripes). This complicates linking SNR levels to true and false positive probabilities. Instead, we later establish potential true vs. (random) false positives by inspecting independent subsets of the campaign data.

PAGE 6

Figure 4, last sentence—needs citation

Citations (45, 46) have been added.

Figure 5—When you say “no prior constraints were assumed for semi-major axis”, I assume the figure still encapsulates the unknown inclination angle of the system and the fact that a planet on an inclined orbit would be harder to detect? It looks like your text says that, but you might want to add a sentence to the caption.

We have clarified “no prior constraints” to say “radius and semi-major axis were uniformly sampled along with an inclination prior of $P(i) \propto \sin i$ ”. While the binary’s orbital inclination is indeed known, it’s possible (perhaps arguably unlikely) that the planetary system could be misaligned. We also tested restricting the plane of the planetary orbits to the orbital plane of the binary, and found that this did not significantly alter the completeness maps. This is noted and highlighted in the Completeness Analysis section of the Supplementary Methods.

Figure 5—Related to the previous point, you should assume some sort of a prior on inclination.

Either something related to the orbit of the binary, or at a minimum, $P(i)$ is proportional to $\sin(i)$ for random sky orientations.

Indeed, we assumed $P(i) \propto \sin i$, and have highlighted in the Supplementary Methods and figure caption where this is indicated.

PAGE 7

Line 23—give the radius (or range of radii) of the putative giant planet.

The plausible range of giant planet radii corresponding approximately from sub-Neptune to Jupiter-sized planets ($R \sim 3\text{--}11 R_{\oplus}$) has been added.

SUPPLEMENT:

When doing background subtraction, how do you compensate for Alpha Cen A and B being different brightnesses?

The following has been added to the Data Reduction and Processing section:

"No scaling was performed to normalize the PSFs, as the purpose of chop subtraction is primarily to remove the ELFN and residual background structure such as the AGPM glow. The residual coronagraphic PSF is also partially mitigated by chop subtraction."

If the planetary systems are aligned with the binary orbit then the masked region of Figure 2 is precisely where you would expect to detect planets. Luckily this does not appear to be the case. You may want to reference some of the recent ALMA work that discussed the co-alignment of binary systems.

This point is complicated by the fact that the orbital plane of the binary is not aligned with their current adjoining line. We utilized the orbital solution in Kervella et al. 2016 (reference 35) to produce the following plot:

If the plane of the orbits is restricted to the orbital plane of the binary (oriented roughly SW to NE), this would fortunately not be among the regions most impacted by detector persistence (roughly SE to NW, compare to Fig. 1). We also tested restricting the plane of the planetary orbits to the orbital plane of the binary, and found that this did not significantly alter the completeness maps in Figure 4. Finally, we note that C1 (located 0.9" to the SW of A) is in the direction of the long axis of the sky projected A/B orbit. This does not prove anything for C1's possible orbit, but would be the most likely position for a detection assuming a planetary orbit with the same inclination as the A/B orbit.

Reviewer #2 (Remarks to the Author):

The authors present a novel observational effort to directly image low-mass planets in the nearest stellar system, alpha Centauri. The team has invested an unprecedented amount of time at the Very Large Telescope for these efforts—77 hours of on-source integration time over the course of several weeks—which is nearly two order of magnitude longer than the typical integration time used in traditional high-contrast imaging surveys. The observations were carried out at mid-infrared wavelengths (10 microns) where thermal emission from low-mass planets is highest. This is challenging because Earth's atmosphere and telescope optical surfaces are also bright in this region, requiring unique technical setup and considerations. Moreover, alpha Cen is a modest-separation binary (with a wide M dwarf tertiary), so a creative observing and processing strategy was needed.

As a feasibility study to explore new parameter space (both in general and around these stars), this seems to have been a success. With reasonable assumptions about planet albedo, thermal evolution, and internal heating, this experiment appears to have reached sub-Saturn sensitivities throughout much of the classically-defined habitable zone of alpha Cen, and down to Neptune sized planets in the most sensitive region—the best mass sensitivity limits with direct imaging for any system, as far as I am aware. This is impressive and in many ways transformational compared to the standard approach to direct imaging surveys over the past two decades, which has been limited to Jovian-mass planets around young stars. This also serves as a preview of future possibilities in the mid-infrared with the upcoming thirty-meter class telescopes and, soon, JWST.

My primary concern with this study relates to the interpretation of the emission feature located near the inner edge of the habitable zone and which the authors attribute to a potential planet or warm circumstellar dust. The authors have carried out a series of experiments to gauge the significance and robustness of this feature. They arrive at a SNR of 3.1-3.5. The residuals after PSF subtraction in high-contrast imaging datasets are plagued by non-Gaussian noise statistics as well as systematics from imperfect AO correction, so any detection with a SNR less than ~5 should be treated with extreme caution. This is mitigated at some level by the long wavelengths and high Strehl ratios in this study (which should be quoted). The signal also appears to be present after splitting the campaign data in half, and after processing even-odd nights (although no significance level is given for these, as far as I can tell, but presumably they have $\text{SNR} < 3$).

However, there are many sources of systematics like persistence stripes and ghosts in this dataset. Other apparent sources appear to be present in the processed images near the circular ring marking the habitable zone inner edge but before the wide strip covering the persistence stripes is masked out. These are evident in Figure S1 (B, E, and F panels) and Figure S3 (A and B panels) at ~11 o'clock and ~2 o'clock. Given the low SNR of the detection and the other artifacts in the image, it is not clear to me that the "C1" feature is any different.

We share the reviewer's opinion that the sensitivity limits are the most important and robust of the results. We also share this reviewer's primary concern: because we cannot be certain that the "C1" feature is an astrophysical detection, and not a systematic artifact, we have left its detection status in candidacy. Indeed, the significance of the source in the final image is just above $\text{SNR} \sim 3$. We furthermore share the reviewer's concern about high-contrast imaging detections (in single epochs) with $\text{SNR} < 5$ as having a higher than expected chance of being due to random (speckle) noise compared to similar significance levels in datasets governed by Gaussian statistics. In this case, the statistics are certainly non-Gaussian due to the level of systematic structure in the images. Instead, our confidence in the positive nature of the C1 detection (either as an astrophysical source, or as a systematic feature) comes from its repeatability in multiple independent subsets of the campaign data, which have $\text{SNR} \sim 2.4\text{--}2.8$ (now indicated in Figure S2). Certainly, this would also be expected of a systematic feature; however, we are unaware of any that would cause a point-like source that rotates with the field of view. This is not true of the other artifacts in the image, including those indicated in Figures S1 and S3, which can each be attributed to a known systematic source (persistence within the Aquarius detector, optical ghosts from the filter, etc.). C1 is the only source in the images that we cannot attribute to any known systematics or to random noise. Therefore, we report it as a candidate exoplanet or exozodiacal disk. This will certainly require confirmation, which unfortunately may be challenging and could take considerable time due to the inordinate resources that must necessarily be devoted toward such an observation. We hope by publishing this candidate now that the astronomical community will be inspired to confirm or reject these three hypotheses.

The typical Strehl ratio of $>97\%$ (measured by the AO wavefront error and extrapolated to the N-band using Marechal's equation) has been added to the Instrumental Setup and Observing Strategy section of the Supplementary Methods.

Other minor comments:

(1) The integration time of the data that were used was 77 hours, not 100 hours. 100 hours should be removed from the abstract.

The reviewer is correct that only 77 hours of data were used in the final analysis. However, we intended to quote the open shutter time vs. only the data passing various quality checks in order to not be overly optimistic of the expenditure of telescope time required for the

experiment. We have changed the word “100 hours of observations” to “100 hours of open shutter time” to clarify this distinction.

(2) Please include how habitable zone is being defined, as this can dramatically vary depending on assumptions about a planet’s atmospheric pressure and greenhouse effects.

We have clarified in the first paragraph of the main text that in this context “habitable” refers to the possibility of a planet with an Earth-like atmosphere to host liquid water on its surface.

(3) There is a substantial difference (by ~ 2 orders of magnitude!) between the 3 sigma pixel noise contrast curve and the 3 sigma injection-recovery contrast curve in Figure 4, especially at small angular separations. The injection-recovery approach should be more robust as it mimics the practice of actually detecting point sources. Because of the large discrepancy between the two approaches, the pixel noise curve should not be displayed in this figure.

On the one hand, we believe that this curve represents an important result that illustrates the fundamental limitations of the current detector technologies. However, on the other hand, we also share the reviewer’s concern that this might be errantly interpreted as a point-source detection limit to be compared to the injection-recovery tests. As a compromise, we have indicated in the figure caption that the “injection/recovery” curve corresponds to the sensitivity to point-sources, and have indicated that the dashed curve corresponds to the “background noise contribution from the variation of pixel-pixel intensities”. This furthermore serves as a useful illustration of the various noise contributions (see comments by the other reviewers). The difference is a factor of ~ 4 in the high-contrast region at separations larger than ~ 1 , while larger discrepancies are mostly at separations $\leq 2 \lambda/D$. The dashed line represents the ultimate contrast limit our observations could have reached with a perfect correction of PSF residuals by the coronagraph and the AO.

(4) Please explain where the Bond albedo of 0.3 comes from (presumably Earth).

We have indicated in the expanded Simulated Planetary Contrast vs. Separations Curves section that the Bond albedo of 0.3 comes from assumptions that the planets are similar to Earth. Uranus, Neptune, Saturn, and Jupiter also have similar Bond albedos (see <https://nssdc.gsfc.nasa.gov/planetary/factsheet/>). We have also added reference 60 for this point. As further justification for the singular choice of values for this parameter, we note that the planetary equilibrium temperatures are a weak function of the albedo (proportional to the albedo to the 0.25 power).

Reviewer #3 (Remarks to the Author):

This is an intriguing paper. It presents some novel aspects of IR coronagraphic imaging of alpha Cen A and B and reports a possible object in orbit around A. The work is excellent, but lacks some important aspects in the analysis of the candidate they call C1.

We appreciate the reviewer's optimism and complementary remarks of our study. We also agree on the importance of the possible object in orbit around Alpha Centauri A, and the necessity for a thorough analysis. We further appreciate the reviewer's thoughtful ideas to improve upon our original analysis, and have attempted to closely follow each of their suggestions.

The Technique

The authors present a novel way of using chopping in 10 micron, ground based imaging by exploiting a chopping secondary (actually a deformable mirror). The technique is sound and in principle could be replicated elsewhere (with significant investment by other observatories).

The other novelty is using the fact that A and B are only 15 arcseconds apart, so they partially serve as PSF reference stars for each other in the image data reduction scheme. This is quite clever, and probably can be improved upon with some other tricks, and can be used in other attempts to image substellar objects in binary systems, an area that is ripe for discoveries, but has been technically difficult, although methods are being studied by others as well.

We agree that binary differential imaging (e.g., Kasper et al. 2007, Rodigas et al. 2015) is in principle a powerful method of obtaining reference PSFs. In practice, this is limited by the size of the isoplanatic angle (on the scale of a few arcseconds for ground-based imaging). For the purposes of NEAR, the primary reason of the chop subtraction was to remove the background and detector excess noise. A fortunate side effect is that the chop subtraction partially subtracts the residual coronagraphic PSF of Alpha Cen B from Alpha Cen A, leading to a contrast improvement of about a factor of two (B is about half as bright as A in the N-band). An even more significant gain would be expected if both stars were equally bright. We include these details for the relevance of this discussion, but have not added these tangential points to the manuscript. As a side note, the separation of A and B in 2019 was 5.15" (see reference 35: Kervella et al. 2016, and later responses).

The authors argue that with their coronagraph and the PSF removal, followed by some artifact removal and pesky detector persistence, they would be able to image planets as small as Saturn, or maybe even super-earth sized ones around these two stars, if they had roughly 100 hours of observing time. This was not entirely convincing.

We regret that the sensitivity analysis, which we believe is the most important result of the study, was not entirely convincing. After reading the following suggestion, we understand that this was due to the lack of detail given for our planetary brightness models. We hope

that after a complete rewrite and expansion of this section, including additional details and supporting references and equations, that these models are more easily understood and reproducible by the general reader. Furthermore, we have compared the results of our models to the brightness of Earth, and have found that our models yield consistent results.

First, the sensitivity analysis uses a very crude model for the brightness of these simulated planets, which is basically a black body function plus some chosen albedo, although, even with the supplemental material, the model is not reproducible by a reader. They try several variations on the internal heating of their putative planets. However, we know for a fact that Earth, Jupiter, Neptune etc are not black bodies and in fact at these wavelengths there are many (quite interesting, actually) molecular absorption features that deviate the spectral energy distribution of the emergent spectra considerably from a black body function. The main issue here is that this is the fundamental basis for their argument that given 100 hours of VLT time to do this experiment again, they could detect such things. Everything hinges on the model flux density in the N-band (~10 μ m). The description, even in the supplemental material, is decidedly vague and does not even give what temperatures were used for each object or what radii were used, what irradiance they used, etc. Regardless, the model is so simplistic that it evokes significant doubt in the results. In addition, simply amping up internal heat sources would have a significant effect on the atmospheres of these things as well as their emergent spectra.

The reviewer is correct that our description of the spectral models was too sparse to facilitate reproducibility. This was unintentional, and we are grateful for the suggestion to improve upon our description. We have completely re-written this section (previously a single paragraph) into a full-page description of the model. Indeed, the model is relatively simplistic. However, we see this as an advantage given the broad photometric bandpass and dominant source of uncertainty that is manifest in the level of additional heating. We copy only a portion of this re-written section here for the purposes of this discussion:

“For Earth’s atmosphere, the dominant absorbers in the λ ~8–13 μ m range are O₃ at λ < 10 μ m and CO₂ at λ > 12.5 μ m (61). Therefore, the overall level of absorption for Earth-like planets is expected to be low within the λ = 10–12.5 μ m range of the NEAR filter (14). For Neptune, the dominant feature within this range is C₂H₆ emission at 12.2 μ m, which is an order of magnitude above the thermal continuum (62). For Jupiter, NH₃ absorption at ~11 μ m is present at a comparable equivalent width to the also present C₂H₆ emission (63). Such planetary atmospheres at ~1 au would have comparable equilibrium temperatures to Earth, which will change the overall ratios of these constituents and their spectral contributions. However, more detailed modeling of such atmospheres also suggests that blackbody spectra are reasonable approximations for λ = 10–12.5 μ m (42).”

The Candidate Companion

It is extremely exciting that the team found a candidate companion to Alpha Cen A, which they call C1. Whatever it is, it is an important discovery, if it is orbiting A. It not need be a Jupiter

radius planet to be exciting. For example, brown dwarfs are also about that radius, and finding one in this system would be equally important.

We agree that finding a brown dwarf within Alpha Centauri would also be an important discovery. However, such a companion would appear at least a factor of two brighter than C1 (assuming a radius similar to Jupiter). $M\sin i \geq 53 M_{\oplus}$ companions within the habitable zone α Centauri A are also excluded by existing RV observations (see reference 20, Zhang et al. 2018) for all but the smallest inclinations from face-on ($<0.7^{\circ}$ assuming $M \sim 15 M_{\text{Jup}}$).

However, although the detection of something next to Alpha Cen A seems certain, the arguments that it is orbiting A were again too simplistic. A simple quick calculation suggests that its small motion from May to June is actually quite consistent with the motion of A on the sky, although I am confused by the images presented. A is about ~ 15 arcsec to the NE of B, so it seems that the compass on the image figure may be mistaking the cardinal directions (is N really down, not up as indicated?). It would be nice to label which residuals belong to which star when it is unocculted in Fig. 2A (as done in the reference 25 paper), and it would be helpful if figure 1B were actually oriented the way the system is on the sky. (N up, E to the left, is the convention, although of course this was done in the southern hemisphere.)

We are less confident than the reviewer that C1 is an astrophysical detection. Although we know of no artifact that would mimic the properties of C1 (a point-source that rotates with the parallactic angle), given the number of artifacts within the image, we cannot be certain that C1 is not an unknown systematic artifact. However, we agree with the reviewer on the importance of this potential discovery, which is our reason for publishing C1 while it is still in candidate status.

Regarding the motions of the candidate: there was no statistically significant motion that was observed. All astrometric measurements were within ~ 1 - 2 -sigma of their mean (amounting to ~ 100 - 200 mas at most). We've also provided a check that orbital motion would not produce a larger astrometric shift, but we do not claim to have confidently detected motion in C1.

The images are displayed with the correct orientation. North is indeed up, and East is indeed left, as indicated by the compass. We are not sure where the measurement of A being ~ 15 arcsec to the NE of B originated from, but it is not correct for the stars in 2019. In June 2019, Alpha Cen B was located 5.15 arcsec away from A to the NW (specifically at -18.6° W of N, see reference 43: Kervella et al. 2016). Similarly, Figure 1B is orientated in an approximate N up, E left orientation, with slight differences to facilitate illustration of the system's habitable zones and orbital parameters. The off-axis PSFs have been labeled in Figure 2, which should also help to alleviate potential confusion.

Even if the orientation of the images is correct, it is important to consider the motion of A over the period of observations. Three different dates of observations were separated by 2.5 and 2

weeks respectively (roughly). A has a proper motion of ~ 170 mas South and ~ 33 mas E in 2.5 weeks ($-3608, 686$)mas/yr. This is actually consistent with the delta position mentioned in the supplementary material (if those directions are correct). However, on top of this, A also has a parallax of 742 mas. In 2.5 weeks, the parallactic motion then is $2 \times 35.7 \text{ mas} = 71$ mas in two weeks (actually more, since it is so far south and its parallactic motion will be nearly circular, but this is just a rough estimate). Whether these cancel each other out or actually magnify each other, the authors need to consider these issues, and a plot of the motion on the sky would be helpful. There is also the third component which is the motion around the AB barycenter. It is possible that doing this analysis properly makes the case for C1 to be a candidate even much stronger. The best and only certain method of companion confirmation is common parallax, which requires gravitational interaction. So at some level, the authors may have a more secure result than they know, although this is complex and has to be analyzed properly, including all of the motions involved.

As mentioned in a previous comment, the astrometric measurements are consistent with no motion. While a slight motion at the ~ 1 - 2 sigma level was measured, we are not confident that this is an actual observed motion. We agree with the reviewer's assessment of the system's motions, and that the level (or lack) of observed motion of C1 is consistent with a background motion over this short timeframe. However, as noted, these measurements are also consistent with a planetary orbit, static exozodiacal disk, and static instrumental artifact. We ultimately determined not to include the plot of the system's motions in the manuscript because it does not add any new or unpublished information (however, see our response to the next comment, which does include such a plot of the system's motions).

I did not find the pre-imaging argument using K band data in 2009 convincing. It could be with some figures, perhaps, showing how A is moving around over that period of time (i.e. not just proper motion).

We discussed whether the pre-imaging argument would be convincing without replicating the images published in reference 43. Since an interested reader can find this already published information, we determined that duplication was not warranted. To facilitate the present discussion, we reproduce the *Ks*-band version of the relevant image in this review document and refer to 43 (Kervella+2016, A&A 594, 107, Fig. 6) for details on the observations and data reduction. The following image contains the astrometric tracks (including the proper motion, parallax, and orbit) of Alpha Cen A and B throughout 2017–2021. This *Ks*-band ($\sim 2 \mu\text{m}$) image shows a source that is ~ 2 x fainter than C1 near the trace of Alpha Cen B in 2021. If C1 were a background star, it would be clearly visible in this image near the trace of Alpha Cen A (orange-yellow track) in 2019, as indicated by our overlaid blue circle. A background galaxy is furthermore highly unlikely (see Supplementary Methods).

Finally, there are many examples of claims of finding companions that later turned out to be irreproducible or incorrect, something I am sure the authors are aware of. It is a bit of a warning sign that this thing was not detected in the third data set, although certainly the argument that it moved into the bad part of the field of view is possibly true. It would be wise for the authors to obtain at least a few more similar data sets before arguing (as they do) that this C1 must be in orbit around A.

We are also aware of these many examples. Indeed, earlier errant claims of companions have resulted in an abundance of caution that has prevented us from claiming that C1 is a bona fide or even likely companion. We are careful to refer to C1 as a “candidate detection”, as we are not confident that it is of astrophysical nature. Regarding the third dataset, comprising only a single night, its exposure time is less than 10% of the larger campaign, and thus the lack of detection on that night does not carry the same significance as the other subsets of campaign data. As noted by the reviewer, this non-detection can also be explained if C1 has indeed orbited outside of the FoV. However, we are similarly not confident in this interpretation, and merely state that it is consistent with the observations.

We also would like to stress an important point here: *we do not claim that C1 must be an object in orbit around Alpha Centauri A.* We agree that additional datasets of similar depth, or a completely independent experiment (such as precision radial velocities or stellar astrometry) would be necessary to make such a claim with confidence. Unfortunately, a

second 100-hr dedication of VLT time is not presently foreseen, although it would be possible (and perhaps highly worthwhile) to repeat the NEAR experiment. Regarding other facilities, JWST could possibly perform an independent high-contrast imaging experiment in several years. Precision astrometry from dedicated space missions that are being planned (e.g., TOLIMAN: Bendek et al. 2018, SPIE, 10698), would also enable an independent experiment to be conducted. For the time being, there is not an available path forward for our team to obtain additional data to verify the nature of C1. That is why we have chosen to publish its discovery as a potential candidate, as other teams may reach such a verification earlier or via other means than we have anticipated. If not published, then opportunities for confirmation may be overlooked or altogether missed.

General Comments:

I found the paper to be written stylistically like a proposal to the VLT Time Allocation Committee to get 100 hours of telescope time. That is fine, but the authors should be aware of that impression. However, I hope these comments in this review help the authors to make this a much stronger paper. It was certainly exciting to read, but it needs more work.

We did not intend for the paper to read like a telescope proposal and we appreciate this communication. Our intention was rather to report the results of the already completed 100 hour observational campaign. The time commitment was mentioned repeatedly because this is a novel aspect of the study. No other exoplanet imaging observation with such an exposure time has yet been performed. This impression might also be due to the fact that the NEAR experiment is a pathfinder for ELT instruments like METIS, and we found it to be important to present NEAR's performance and results in that context. We hope that our changes in response to the reviewer's thoughtful and constructive comments have helped to adjust this perceived tone for future readers.

REVIEWERS' COMMENTS

Reviewer #1 (Remarks to the Author):

Thanks for addressing my comments. Your new section discussing the sensitivity limit and how it extrapolates to Earth-imaging with the E-ELT greatly strengthens the manuscript.

-Andy Skemer

Reviewer #2 (Remarks to the Author):

I appreciate the changes the authors made to the manuscript and their response to my concerns.

As indicated in the responses to the referees, C1 is either an exoplanet candidate, exo-zodi disk, or static instrumental artifact of unknown origin. Since this last option is still a serious possibility, I think this needs to be made clearer in the main text. I therefore suggest adding this possibility to the sentence "Therefore, we consider C1 to be a plausible exoplanet and/or exozodiacal disk candidate" (bottom of the third paragraph from the end).

I also think the SNR of the detection should be quoted up front in the main text. Currently, when reading the main text, it's not clear whether the SNR could be 2 or 20.

Regarding the total open shutter time of 100 hours listed in the abstract: the changes the authors made from "100 hours of observations" to "100 hours of open shutter time" don't sound very different to me. Instead, to clarify that the detection was not made by integrating for 100 hours, but to retain that cumulative size of the campaign, I suggest language like "Based on the best quality images from 100 hours of cumulative observations, we demonstrate..."

Reviewer #3 (Remarks to the Author):

I believe the authors addressed all the issues sufficiently and I recommend publication.

REVIEWERS' SUBSEQUENT COMMENTS

Reviewer #1 (Remarks to the Author):

Thanks for addressing my comments. Your new section discussing the sensitivity limit and how it extrapolates to Earth-imaging with the E-ELT greatly strengthens the manuscript.

-Andy Skemer

We agree that this discussion strengthens the manuscript, and are grateful for this helpful suggestion, as well Dr. Skemer's other insightful comments.

Reviewer #2 (Remarks to the Author):

I appreciate the changes the authors made to the manuscript and their response to my concerns.

As indicated in the responses to the referees, C1 is either an exoplanet candidate, exo-zodi disk, or static instrumental artifact of unknown origin. Since this last option is still a serious possibility, I think this needs to be made clearer in the main text. I therefore suggest adding this possibility to the sentence "Therefore, we consider C1 to be a plausible exoplanet and/or exozodiacal disk candidate" (bottom of the third paragraph from the end).

We agree with Reviewer #2 that the possibility that C1 could be an (unknown) systematic artifact should be very clear in the manuscript. We have added the following sentence after the one indicated:

"While C1 cannot be explained by presently known systematic artifacts, an independent experiment is necessary to exclude this third possibility."

I also think the SNR of the detection should be quoted up front in the main text. Currently, when reading the main text, it's not clear whether the SNR could be 2 or 20.

This is another good point. The low SNR of C1 is an important aspect that should be easily understood from the main text. We have modified the following sentence that serves to introduce C1:

"In a relatively clean region of the image, there is one point-like feature (signal to noise ratio ~ 3) that is not associated with any known detector artifacts."

Regarding the total open shutter time of 100 hours listed in the abstract: the changes the authors made from "100 hours of observations" to "100 hours of open shutter time" don't sound very different to me. Instead, to clarify that the detection was not made by integrating for 100 hours,

but to retain that cumulative size of the campaign, I suggest language like “Based on the best quality images from 100 hours of cumulative observations, we demonstrate...”

We agree that the wording suggested by Reviewer #3 more accurately describes the exposure time. We have slightly modified the suggested wording in the following revised sentence:

“Based on 75–80% of the best quality images from 100 hours of cumulative observations, we demonstrate...”

Reviewer #3 (Remarks to the Author):

I believe the authors addressed all the issues sufficiently and I recommend publication.

We would like to thank Reviewer #3 again for their time and insight in the review process. As the editors have pointed out, the discussions regarding C1 in response to their comments are a useful addendum to the discussions in the main text.